# Dysregulation of AMPA Receptor Trafficking and Intracellular Vesicular Sorting in the Prefrontal Cortex of Dopamine Transporter Knock-Out Rats

**DOI:** 10.3390/biom13030516

**Published:** 2023-03-11

**Authors:** Giorgia Targa, Francesca Mottarlini, Beatrice Rizzi, Damiana Leo, Lucia Caffino, Fabio Fumagalli

**Affiliations:** 1Department of Pharmacological and Biomolecular Sciences “Rodolfo Paoletti”, Università degli Studi di Milano, Via Balzaretti 9, 20133 Milano, Italy; 2Department of Neurosciences, University of Mons, 6 Avenue du Champ de Mars, 7000 Mons, Belgium

**Keywords:** glutamate, prefrontal cortex, rat, dopamine transporter

## Abstract

Dopamine (DA) and glutamate interact, influencing neural excitability and promoting synaptic plasticity. However, little is known regarding the molecular mechanisms underlying this crosstalk. Since perturbation of DA-AMPA receptor interaction might sustain pathological conditions, the major aim of our work was to evaluate the effect of the hyperactive DA system on the AMPA subunit composition, trafficking, and membrane localization in the prefrontal cortex (PFC). Taking advantage of dopamine transporter knock-out (DAT^−/−^) rats, we found that DA overactivity reduced the translation of cortical AMPA receptors and their localization at both synaptic and extra-synaptic sites through, at least in part, altered intracellular vesicular sorting. Moreover, the reduced expression of AMPA receptor-specific anchoring proteins and structural markers, such as Neuroligin-1 and nCadherin, likely indicate a pattern of synaptic instability. Overall, these data reveal that a condition of hyperdopaminergia markedly alters the homeostatic plasticity of AMPA receptors, suggesting a general destabilization and depotentiation of the AMPA-mediated glutamatergic neurotransmission in the PFC. This effect might be functionally relevant for disorders characterized by elevated dopaminergic activity.

## 1. Introduction

Dopamine (DA) neurotransmission in the prefrontal cortex (PFC) mediates a variety of adaptive behaviors essential for survival, such as goal-oriented behavior, reward learning, cognitive flexibility, and executive functions [1,2,3,4]. Among the neuromodulatory roles of DA in the PFC, it has been widely demonstrated that it modulates and integrates glutamatergic synaptic transmission and differently impacts the excitability of pyramidal neurons through activation of D1 and D2 dopaminergic receptors [5,6,7]. Most pyramidal neurons in the PFC express-amino-3-hydroxy-5-methyl-4-isoxazole propionic acid (AMPA) receptors and are innervated by dopaminergic fibers [8,9]. Moreover, it has been recently demonstrated that in vivo pharmacological manipulations of DA receptors modulate local glutamatergic transmission via AMPA receptor phosphorylation [10]. Indeed, DA and glutamate systems have been shown to bidirectionally communicate, and interference with this interaction underlies several psychiatric disorders, such as addiction, schizophrenia, depression, post-traumatic stress disorder, and attention deficit hyperactivity disorder [5,11,12,13,14,15].

At the synapse, AMPA receptors mediate the majority of fast excitatory neurotransmissions in the brain, depending on their number, localization, and subunit composition. In fact, the amount and composition of AMPA receptors within the postsynaptic density (PSD) of dendritic spines determine synaptic efficacy and affect neuron excitability [16]. In addition, AMPA trafficking to, at, and from the synaptic membrane and their lateral diffusion dynamically regulate the strength of synaptic plasticity [17] and contribute to homeostatic and Hebbian plasticity, such as synaptic scaling, long-term potentiation, and depression [18,19]. Moreover, mechanisms of exo/endocytic traffic between intracellular pools and the cell surface, lateral diffusion along the membrane, and exchange between synaptic and extra-synaptic sites, as well as retention at the PSD, require a variety of scaffolding proteins and cell adhesion molecules that interact with AMPA receptors and drive activity-dependent changes in neuronal circuits and remodeling of synaptic contacts [18]. Changes in AMPA receptor density occur through internalization, a process that, following endocytosis, sorts AMPA receptors between recycling and degradative pathways, depending on the stimulus [20,21]. Neuronal overexcitation up-regulates the activity-regulated cytoskeleton (Arc, also known as Arg 3.1) protein levels that trigger AMPA receptor internalization via its interaction with endocytic machinery [22,23]. The AMPA receptor internalization process couples the cargo into early endosome via the activity of a small monomeric GTPase Rab5 [24], where it should be sorted into distinct pathways. AMPA receptors in the early endosome might be recycled back to the plasma membrane or mature into a late endosome, a mechanism regulated at least in part by Rab9. Further, via the lysosomal pathway through the lipidation of LC3 protein, AMPA receptors may undergo autophagy-dependent degradation [25] or be targeted back to the Golgi, the biosynthetic machinery that confers further post-translational modifications [26].

Since perturbation of DA activity, as under repeated cocaine exposure, impacts cortical AMPA plasticity, thus leading to altered modulation of AMPA receptor subunit composition and trafficking [27,28,29,30], alteration in DA-AMPA crosstalk might contribute to an abnormal functional and architectural synaptic remodeling. Although several lines of evidence exist showing DA-glutamate interaction at the cortical level, little is known about the contribution of DA in the delivery of AMPA receptors in and out of the PSD surface. To mimic a stable condition of DA overactivity, we took advantage of dopamine transporter knock-out (DAT^−/−^) rats known to display a persistently increased dopaminergic transmission [31]. DAT belongs to a family of plasma membrane transporters of solute carrier family 6 (SLC6), and it is responsible for re-uptaking DA back to the presynaptic terminal, thus limiting the duration of DA-mediated neurotransmission and DA signaling intensity, as well as maintaining filled DA stores [32]. In addition, it is now widely demonstrated that it represents a gate/target for neurotoxins and psychostimulants [33,34]. Further, we have recently demonstrated that the removal of DAT leads to a working memory deficit coupled with alterations of neuroplastic molecules, such as the Brain-Derived Neurotrophic Factor (BDNF) [31]. In that manuscript, we found that BDNF expression in the PFC is reduced in the PSD of DAT^−/−^ rats, an observation that may support the hypothesis that DA overactivity might control the homeostasis of AMPA-mediated glutamate neurotransmission, since it is known that BDNF regulates glutamatergic functionality [35]. Based on the evidence mentioned above, our goal was to investigate if and how hyperactivity of the DAergic system at the prefrontal cortex level, shown by previous work in DAT^−/−^ mice [36], would influence AMPA subunit composition, trafficking, and localization. Analyses have been undertaken in the whole homogenate, which gives us information about translational changes in the PSD, informing us about the synaptic localization of receptors and, at the extra-synaptic level, providing clues about the shuttling of these receptors between synaptic and extra-synaptic sites. We also investigated the expression of the main scaffolding proteins of these receptors and structural proteins in the PSD, since they contribute to anchoring them at the membrane, thus allowing physiological neurotransmission. Further, we focused our attention on some endosomal and autophagic mechanisms that may contribute to regulating AMPA endosomal sorting.

## 2. Materials and Methods

### 2.1. Animals and Housing

Zinc-finger nuclease (ZNF) design, construction, in vitro validation, microinjection, and founder selection were performed, as previously described [37]. The target site for ZFN was CTCATCAACCCGCCACAGAcaccaGTGGAGGCTCAAGAG in Exon 2 of the Slc6a3 gene (NCBI Gene ID: 24898; Genomic NCBI Ref Seq: NC_005100.3; mRNA NCBI Ref Seq: NM_012694.2). The dopamine transporter (DAT) knocked-out (DAT^−/−^) lines were created in the outbred Wistar Han background at SAGE Labs. Animals were housed, divided by their respective gender, in groups of three to four with water and food ad libitum. Rats were kept under standard conditions at 22 °C and on a 12 h light/dark cycle (light from 07:00 h to 19:00 h). All experiments were conducted following the guidelines established by the European Community Council (Directive 2010/63/EU of 22 September 2010) and were approved by the Belgian Ministry of Health (Neurosciences LA1500024). All efforts were pursued to minimize animal suffering and to reduce the number of animals used in the experiment.

### 2.2. Protein Extract Preparation and Western Blot Analysis

Adult male rats, both Wistar wild-type DAT^+/+^ (n = 6) and DAT^−/−^ (n = 6), were sacrificed by decapitation, brains were collected, and, immediately after, the medial prefrontal cortices (mPFC) were dissected from 2 mm thick slices in accordance with the Rat Brain Atlas of Paxinos and Watson [38]. Then, PFCs were frozen on dry ice and stored at −80 °C for upcoming molecular analysis.

Proteins in the whole homogenate, post-synaptic, and extra-synaptic fractions were analyzed, as previously described [39]. Proteins from medial prefrontal cortex (mPFC) tissues were homogenized in a Teflon-glass potter using a cold buffer pH 7.4, containing 0.32 M sucrose, 0.1 mM PMSF, 1 mM HEPES, 1 mM MgCl_2_, and 1 mM NaHCO_3_ in the presence of commercial cocktails of protease (cOmplete™ Protease Inhibitor Cocktail, Roche, Monza, Italy) and phosphatase (Sigma-Aldrich, Milan, Italy) inhibitors. An aliquot of the whole homogenate was kept, sonicated, and stored at −20 °C. The remaining homogenate was centrifuged at 800× *g* for 5 min, and the resulting supernatant was subsequently centrifuged at 13,000× *g* for 15 min. The pellet obtained was resuspended in a buffer containing 75 mM KCl and 1% Triton X-100 and centrifuged at 100,000× *g* for 1 h. The resulting supernatant, also called Triton X-100 soluble fraction (TSF, extra-synaptic fraction), was stored at −20 °C, while the pellet, also called Triton X-100 insoluble fraction (TIF, post-synaptic density fraction), was homogenized in a glass–glass potter in 20 mM HEPES, glycerol 30%, protease and phosphatase inhibitors, and stored at −20 °C. The total protein amount was measured in the homogenate, TIF, and TSF according to the Bradford Protein Assay kit from Bio-Rad (Milan, Italy), using bovine serum albumin as the calibration standard. Identical quantities of proteins from homogenate (8 mg), TIF fraction (8 mg), and TSF fraction (35 mg) were run on a sodium dodecyl sulfate 8% or 14% polyacrylamide gel under reducing conditions, and they were next transferred into a nitrocellulose membrane (GE Healthcare, Milan, Italy). Membranes were cut to allow the simultaneous detection of more proteins on one gel; then, blots were blocked for 1 h at room temperature (RT) with I-Block solution (Life Technologies Italia, Monza, Italy) in TBS 0.1% Tween-20 buffer and incubated with antibodies against proteins of interest. To allow the detection of more determinants on the same blot, the latter was stripped, blocked, and re-incubated with another antibody.

The conditions of the primary antibodies were the following: anti-vGluT1 (1:1000, Cell Signaling Technology, Danvers, MA, USA, cod. 12331, RRID: AB_2797887), anti-GLT-1 (1:2000, Abcam, Boston, MA, USA, cod. Ab41621, RRID: AB_941782), anti-GluA1 (1:1000, Cell Signaling Technology, cod. 13185, RRID: AB_2732897), anti-GluA2 (1:1000, Cell Signaling Technology, cod. 5306, RRID: AB_10622024), anti-GluA3 (1:1000, Millipore, Burlington, MA, USA, cod. MAB5416, RRID: AB_2113897), anti-SAP97 (1:1000, Abcam, cod. Ab69737, RRID: AB_2091910), anti-GRIP (1:1000, Synaptic System, Göttingen, Germany, cod. 151003, RRID: AB_10804287), anti-PSD95 (1:2000, Cell Signaling Technology cod.3450, RRID: AB_2292883), anti-CRMP-2 (1:1000, Cell Signaling Technology, cod. 9393, RRID: AB_2094339), anti-nCadherin (1:2000, Santa Cruz, CA, USA, cod. sc59987, RRID: AB_781744), anti-Neuroligin-1 (1:1000, Synaptic System, cod. 129 003, RRID: AB_887746), anti-Arc/Arg3.1 (1:500, BD Biosciences, San Jose, CA, USA, cod. 612603, RRID: AB_399886), anti-Rab5 (1:2000, Cell Signaling Technology, cod. 2143, RRID: AB_823625), anti-Rab9 (1:2000, Abcam, cod. ab179815, RRID: AB_303323), anti β1,4-galactosyltransferase 1 (GalT) (1:1000, Abcam, cod. ab178406), anti-Golgi Matrix Protein 130 (GM130) (1:2000, Sigma-Aldrich, Milan, Italy, cod. G7295, RRID: AB_532244), anti-LC3B II (1:1000, Invitrogen, Waltham, MA, USA, cod. PA1-16930, RRID: AB_2281384), and anti-β-actin (1:5000, Sigma-Aldrich, cod. A5441, RRID: AB_476744). Results were standardized to β-actin and detected by evaluating the band density at 43 kDa.

Immunocomplexes were acquired by chemiluminescence using the Chemidoc MP Imaging System (Bio-Rad Laboratories, Hercules, CA, USA, RRID: SCR_019037) and analyzed with Image Lab^TM^ software (Bio-Rad, RRID: SCR_014210). The full-size cropped immunoblots are presented in Appendix A. Since gels were run at least two times, the obtained results were averaged with a correction factor: correction factor gel B = average of (OD protein of interest/OD β-actin for each sample loaded in gel A)/(OD protein of interest/OD β-actin for the same sample loaded in gel B) [40].

### 2.3. Statistical Analysis

Data were collected in individual animals as independent determinations, and they are addressed as means ± standard errors.

Molecular changes in protein levels produced by genotype were tested for normality of residuals with the Kolmogorov-Smirnov test. Data with normal distribution were analyzed by unpaired Student’s *t*-test (t), using as the control condition DAT^+/+^ animals and, as the testing condition, DAT^−/−^. Data with a non-normal distribution were analyzed by the Mann-Whitney test (U).

Subjects were eliminated from the final dataset if their data deviated from the mean by 2 SDs. Prism 9 (GraphPad Software Prism v9, San Diego, CA, USA, RRID: SCR_002798) was used to analyze all data. Significance for all tests was assumed at *p* < 0.05.

## 3. Results

We started by analyzing the levels of the vesicular glutamate transporter type 1 (vGluT1), which is responsible for the packaging of glutamate into synaptic vesicles for its exocytotic release [41,42]. Figure 1a shows a significant increase in the expression of vGluT1 in the mPFC of DAT^−/−^ rats (Figure 1a: +26% vs. DAT^+/+^, t = 3.667, *p* = 0.0043). We then analyzed the expression of the glial glutamate transporter, GLT-1, which removes the extracellular glutamate from the synaptic cleft, thus regulating synaptic glutamate levels [43]. Figure 1b shows a significant increase in the expression of GLT-1 (Figure 1b: +20% vs. DAT^+/+^, U = 1, *p* = 0.0043) in the mPFC of DAT^−/−^ animals.

We next investigated the expression of AMPA receptor subunits and their coupled scaffolding proteins [44]. To dissect the effect of DA overactivity on protein translation from their availability at synaptic and extra-synaptic sites, we evaluated the expression of AMPA receptor subunits in the whole homogenate (Homo), synaptic (PSD), and extra-synaptic (Extra-syn) fractions, respectively. We found a significant decrease in GluA1 and GluA2 protein levels in the whole homogenate (Figure 2a: −21% vs. DAT^+/+^, t = 2.511, *p* = 0.0309; Figure 2b: −15% vs. DAT^+/+^, t = 2.582, *p* = 0.0273), whereas GluA3 levels were increased (Figure 2c: +27% vs. DAT^+/+^, t = 2.261, *p* = 0.0473). Analysis of AMPA receptor subunits in the subcellular fractions revealed an overall significant reduction in GluA1 (Figure 2a: PSD: −36% vs. DAT^+/+^, t = 4.267, *p* = 0.0016; Extra-syn: −34% vs. DAT^+/+^, t = 2.720, *p* = 0.0216), GluA2 (Figure 2b: PSD: −22% vs. DAT^+/+^, t = 2.431, *p* = 0.0354; Extra-syn: −15% vs. DAT^+/+^, t = 2.465, *p* = 0.0334), and GluA3 (Figure 2c: PSD: −19% vs. DAT^+/+^, t = 3.616, *p* = 0.0047; Extra-syn: −19% vs. DAT^+/+^, t = 2.395, *p* = 0.0376) subunits in both synaptic and extra-synaptic fractions.

In parallel to AMPA receptor subunits, we also measured the expression of the specific anchoring proteins of GluA1, SAP97, GluA2, and GRIP. Both these scaffolding proteins were reduced in both homogenate and post-synaptic density of the mPFC of DAT^−/−^ rats (SAP97 Figure 3a: Homo: −17% vs. DAT^+/+^, t = 2.485, *p* = 0.0323; PSD: −25% vs. DAT^+/+^, t = 2.362, *p* = 0.0398; GRIP Figure 3b: Homo: −18% vs. DAT^+/+^, t = 3.077, *p* = 0.0117; PSD: −19% vs. DAT^+/+^, t = 2.270, *p* = 0.0477).

To better understand AMPA receptor subunit composition, we measured the GluA1/GluA2 and GluA2/GluA3 ratio in the postsynaptic density. DA overactivity decreased the GluA1/GluA2 ratio (Figure 2d: −17% vs. DAT^+/+^, t = 2.711, *p* = 0.0219), while no changes were observed in the GluA2/GluA3 ratio (Figure 2e: −4% vs. DAT^+/+^, t = 0.5097, *p* = 0.6213).

To investigate whether DA overactivity-induced reduction of AMPA receptor subunits in both synaptic and extra-synaptic fractions might involve endocytic mechanisms, we evaluated Arc/Arg 3.1, a protein that, at the PSD, contributes to AMPA receptor endocytosis [23] and molecular markers of intracellular vesicular trafficking related to AMPA endocytosis and autophagy. Arc/Arg 3.1 protein expression levels were significantly increased in both homogenate and PSD in DAT^−/−^ rats (Figure 4a: Homo: +15% vs. DAT^+/+^, t = 3.255, *p* = 0.0086; PSD: +29% vs. DAT^+/+^, t = 2.721, *p* = 0.0215). Then, we analyzed Rab5, the key endosomal GTPase required for membrane receptors’ internalization and early endosome generation, as well as Rab9, a marker of the late-endosome-to-trans-Golgi-network trafficking [45]. DAT deletion significantly increased Rab5 proteins level (Figure 4b: +31% vs. DAT^+/+^, t = 2.931, *p* = 0.0150), while, in contrast, it reduced Rab9 expression (Figure 4c: −25% vs. DAT^+/+^, t = 4.331, *p* = 0.0015) in the whole homogenate. Next, the trans-Golgi and cis-Golgi markers, GalT and GM130, respectively [46], were reduced in the homogenate of DAT^−/−^ rats (Figure 4d: −25% vs. DAT^+/+^, t = 2.232, *p* = 0.0497; Figure 4e: −37% vs. DAT^+/+^, t = 3.435, *p* = 0.0064). We also measured a well-known autophagy marker, the active form of the light chain protein 3, LC3-II [47], which was significantly decreased in the homogenate of DAT^−/−^ animals (Figure 4f: −40% vs. DAT^+/+^, t = 2.430, *p* = 0.0354).

Finally, to evaluate whether DAT deletion caused structural rearrangements in the mPFC, we evaluated the expression of different structural markers. In particular, we evaluated: (1) PSD95 levels, an integral protein of the post-synaptic density involved in glutamate receptors’ stability [48]; (2) CRMP-2 (collapsing response mediator protein, which is involved in axon formation) [49,50]; (3) nCadherin (nCad), a cell–cell adhesion protein [51] and (4) Neuroligin-1, a protein participating in intercellular junction formation [52]. In the whole homogenate, DA overactivity significantly increased protein levels of PSD95 (Figure 5a: +23% vs. DAT^+/+^, t = 4.119, *p* = 0.0021), nCad (Figure 5c: +29% vs. DAT^+/+^, t = 3.733, *p* = 0.0039), and Neuroligin-1 (Figure 5d: +38% vs. DAT^+/+^, t = 3.269, *p* = 0.0084), whereas CRMP-2 levels were instead reduced (Figure 5b: −21% vs. DAT^+/+^, t = 10.67, *p* < 0.0001). In the post-synaptic density, PSD95 expression was increased (Figure 5a: +51% vs. DAT^+/+^, t = 4.965, *p* = 0.0006). To the contrary, CRMP-2 (Figure 5b: −19% vs. DAT^+/+^, t = 2.713, *p* = 0.0218), nCad (Figure 5c: −24% vs. DAT^+/+^, t = 2.456, *p* = 0.0339), and Neuroligin-1 (Figure 5d: −21% vs. DAT^+/+^, t = 3.471, *p* = 0.0060) levels were decreased.

## 4. Discussion

In this study, we reveal that DA overactivity regulates the translation of cortical AMPA receptors and their localization at synaptic sites through, at least in part, the impairment of the related intracellular vesicular sorting pathway (Figure 6). It appears, thus, that a physiological concentration of DA is critical for the maintenance of AMPA receptor trafficking at the PSD. Our findings represent an aberrant form of neuroplasticity that, via hyperactive ascending DAergic projections to the cortical glutamatergic synapses, might underly working memory and executive function impairments previously observed in both rodents and humans under elevated dopaminergic activity [6,53,54].

First, we found that DA overactivity alters the expression of some critical determinants of the cortical glutamate synapse, i.e., vGluT1 and GLT-1. As suggested by the upregulation of vGluT1, the increase in glutamate release may be buffered by the increased expression of the glial transporter, GLT-1, which may remove the excess of extracellular glutamate and prevent its excitotoxicity. These changes indicate that DA overactivity has caused adaptive rearrangements of the cortical synapse, further strengthening the notion of the tight relationship between DA and glutamate.

More evident changes were observed at the AMPA receptors level, in line with previous findings demonstrating that DA receptors modulate synaptic plasticity by altering AMPA receptors’ expression and surface delivery in cortical neurons [28]. In fact, we demonstrated that DA overactivity dysregulates the glutamate synapse in the mPFC by reducing translation of GluA1 and GluA2, as well as synaptic and extra-synaptic availability of GluA1, GluA2, and GluA3 AMPA receptor subunits. The reduced GluA1 and GluA2 translation might be due to reduced transcription of the different subunits to the modulatory effect of miRNAs or post-translational changes, such as SUMOylation or ubiquitination [55]. Since a physiological communication among DAergic and glutamatergic systems contributes to adaptive behavior fundamental for survival [56], the overall AMPA downregulation herein described suggests a previously undescribed DA-driven maladaptive plasticity that might underly a depotentiation of the post-synaptic strength and responsiveness. Interestingly, our hypothesis aligns with previous data showing that specific DAT inhibitors and elevated DA levels in cortical slices from naïve rats impaired long-term depression, thus inducing abnormal PFC activity [57]. Moreover, it is known that experiences, especially those involving a rise of DA, such as addictive behaviors, are likely to generate or re-program the excitatory synapse toward immature or silent synapse in which AMPA receptors are either absent or highly unstable [58,59], thus regulating drug-associated memories [60]. Moreover, since silent synapses are abundant during the early stage of development [61], we can hypothesize that DA overactivity, via alteration of AMPA abundance and trafficking, might maintain the cortical excitatory synapses in a molecular composition that might prevent synapse maturation and that might remodel cortical neurocircuitry toward a vulnerable endophenotype.

In addition to surface delivery, the distribution of membrane AMPA receptors could be rapidly modified by lateral diffusion via constitutive or regulated pathways [62,63]. The overall reduction in AMPA receptors at both synaptic and extra-synaptic sites, as observed in DAT^−/−^ rats, further corroborates the hypothesis that a hyperdopaminergic state might affect not only AMPA receptor abundance at the membrane, but also AMPA receptor recruitment at the active site of the synapse and the laterally diffusing AMPA receptor pool. Such depotentiation is strengthened by a defective anchoring of these receptors at the post-synaptic membrane, as shown by the reduced expression of their main scaffolding proteins, SAP97 and GRIP, indicating that AMPA receptors are less retained at the PSD of dendritic spines and, thereby, less stable, thus affecting the ability to dynamically remodel the synapse in response to various form of plasticity [19]. Interestingly, such alterations are paralleled by altered synaptic localization of structural markers, such as Neuroligin-1 and nCadherin, which glue together pre- and postsynaptic terminals to sustain physiological neurotransmission [64]. The reduced Neuroligin-1 and nCadherin expression in the PSD further points to unstable synapses and a compromised glutamatergic transmission driven by hyperdopaminergia. Further, the reduced expression and synaptic localization of CRMP-2, a structural marker that facilitates synaptic AMPA receptor trafficking [65], add evidence to the impaired molecular composition of the cortical synapse in DAT^−/−^ rats. Since the reduced localization of Neuroligin-1 at synapse might fail to assemble and anchor the AMPA receptor in the active nanodomain in dendritic spines [66], the increased expression and localization at the PSD of the scaffold protein PSD95 may be interpreted as an attempt to capture labile AMPA receptors [67] or as a maladaptive mechanism that further occludes synaptic transmission [68]. To counteract such synaptic structural instability, the increased translation of Neuroligin-1 and nCadherin, observed in the homogenate of DAT^−/−^ rats, might reflect an adaptive response or an attempt to restore a physiological synaptic communication.

In parallel with the altered trafficking of AMPA receptors, DA overactivity modifies the subunit composition of the remaining receptors in the PSD. In fact, the reduction of the GluA1/GluA2 ratio in DAT^−/−^ rats suggests a less excitable synaptic network, since it is well established that GluA1/GluA2, containing AMPA receptors, are inserted into the PSD as consequence of synaptic plasticity to sustain neuronal activity [69]. In addition, the DA overactivity-induced switch toward GluA2-containing AMPA receptors suggests a weakened synapse and, therefore, indicates the generation of silent synapses, as previously observed [60,70]. Conversely, the unaltered GluA2/GluA3 ratio is indicative of unaffected constitutive recycling at the synaptic level in DAT^−/−^ rats [69], suggesting that DA overactivity primarily affects the activity-dependent trafficking pathway.

Interestingly, when neuronal activity is chronically enhanced, depotentiation of AMPA receptor-mediated transmission occurs via increased internalization of the receptors, a process able to modify the synaptic strength, known as synaptic scaling [18]. Such internalization occurs through exocytic and endocytic events [71] and involves Arc/Arg 3.1. Upon intense synaptic stimulation, Arc/Arg 3.1 rapidly accumulates at the PSD [72], promoting AMPA internalization by interacting with the endocytic machinery [23,73,74]. Accordingly, in parallel with reduced membrane AMPA receptor expression, we observed increased Arc/Arg3.1 levels in the PSD, suggesting that DA hyperactivity regulates homeostatic scaling and shapes synaptic strength by inducing neuroadaptations that resemble the molecular profile of cocaine-exposed rats [75,76].

Another novel finding of the present manuscript derives from the evidence that DA hyperactivity alters the mechanisms subserving endosomal sorting. Interestingly, the increased expression of Rab5, which through a vesicle fusion process regulates AMPA receptor endocytosis and sorting in the early endosomes [24,77], further corroborates the increased AMPA receptor internalization in the mPFC of DAT^−/−^ rats. Following AMPA entry into the endosomal system, Rab5 overexpression generally drives endosomal maturation and increased lysosomal degradation of AMPA-containing vesicles [78]; however, in a condition of DA hyperactivity, the sorting in the late endosome or the fusion in phagosomes are likely to be impaired, as suggested by reduced Rab9 and LC3-II. In addition, we found that DA overactivity reduces the *trans*- and *cis*-Golgi markers, Galt and GM130, respectively, together with a significant reduction of Rab9, a protein that mediates endosome-to-trans-Golgi network transport [79]. Taken together, these results suggest that an altered DA transmission may also affect the recycling pathway that retrogradely targets AMPA receptors back to Golgi outposts [21]. Since depletion of *cis*- and *trans*-Golgi proteins may affect the Golgi ribbon and induce central Golgi fragmentation [46], we can speculate that the defective morphology of the Golgi apparatus, observed at the molecular level, likely contributes to the impaired endosomal traffic in DAT^−/−^ rats. Taken together, such abnormalities in endosomal trafficking may render the glutamate synapse more vulnerable and less responsive to stimuli by promoting the intracellular accumulation of AMPA receptors, a mechanism that closely resembles the impaired endosomal-induced aberrant intracellular accumulation of neurofibrillary tangles of hyper-phosphorylated tau and amyloid-b plaques in Alzheimer’s disease [80] and a-synuclein in Parkinson’s disease [81].

We are aware that our manuscript has some limitations. Although we did not demonstrate a direct causal link between the alteration of AMPA receptor trafficking and vesicular sorting, the changes observed in proteins regulating endosomal sorting sustain our hypothesis. Further, we focused our attention on AMPA receptors, and, therefore, we do not know whether our findings generalize to other types of glutamate receptors (NMDA, kainate, or metabotropic receptors). We are also aware that the norepinephrine transporter may come into play as a functional DAT replacement in the prefrontal cortex of DAT^−/−^ rats [82,83]. However, evidence exists that extracellular levels of DA are significantly enhanced in the PFC of DAT^−/−^ mice [36], potentially ruling out the possibility that NET may have prevented cortical DA hyperactivity. In addition, since DAT functions may also be regulated by protein–protein interactions [84,85], we cannot rule out the possibility that a change in DAT interactor functions may contribute to the alteration of AMPA receptor trafficking at the PSD.

## 5. Conclusions

Taken together, our findings show that DA overactivity alters the homeostatic plasticity of the glutamate synapse in the mPFC through profound dysregulation of AMPA receptor trafficking, an effect likely involving increased internalization of such receptors and alterations in endosomal trafficking and degradation. Our data add further details to the interaction between DA and glutamate neurotransmission, thus reinforcing the hypothesis that an altered communication between these systems likely plays significant roles in shaping maladaptive endophenotypes underlying pathological conditions characterized by enhanced dopaminergic tone.

## Figures and Tables

**Figure 1 biomolecules-13-00516-f001:**
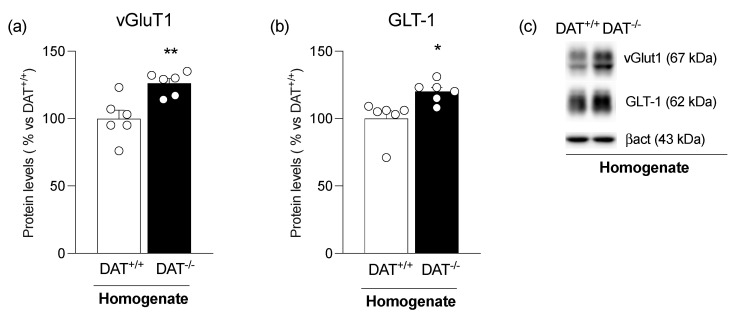
Protein expression level of vGluT1 (**panel a**) and GLT-1 (**panel b**) in the medial prefrontal cortex (mPFC) of DAT^+/+^ and DAT^−/−^ rats. In (**panel c**), representative immunoblots are shown for GLT-1, vGluT1, and β-Actin. Results obtained in the whole homogenate are expressed as mean percentage ± mean standard error from six independent determinations for each experimental group. Unpaired Student’s *t*-test * *p* < 0.05, ** *p* < 0.01 vs. DAT^+/+^ rats.

**Figure 2 biomolecules-13-00516-f002:**
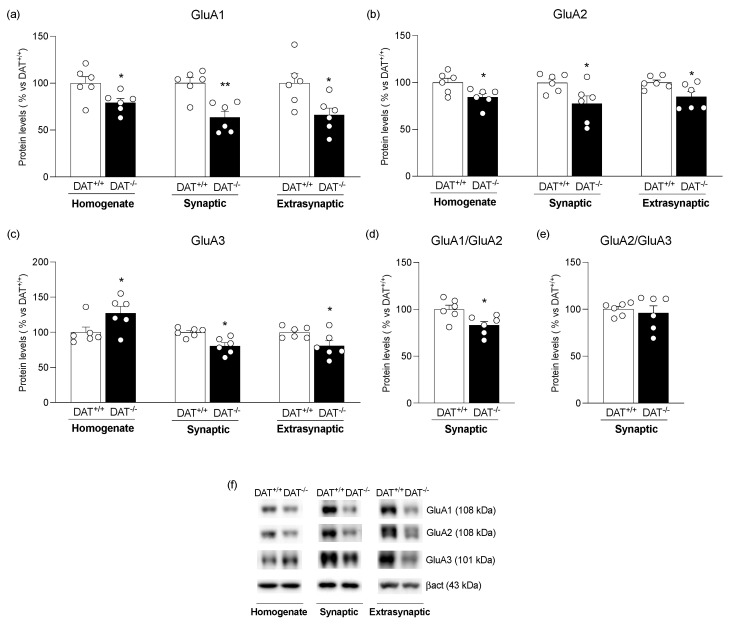
Protein expression level of the main AMPA receptor subunits in the mPFC of DAT^+/+^ and DAT^−/−^ rats. GluA1 (**panel a**), GluA2 (**panel b**), and GluA3 (**panel c**) levels were investigated in the whole homogenate, in the synaptic, and in the extra-synaptic fractions. The ratio GluA1/GluA2 (**panel d**) and GluA2/GluA3 ratio (**panel e**) were measured in the synaptic fraction. In (**panel f**), representative immunoblots for GluA1, GluA2, GluA3, and β-Actin are shown for each fraction evaluated. Results are expressed as mean percentage ± mean standard error from six independent determinations for each experimental group. Unpaired Student’s *t*-test * *p* < 0.05, ** *p* < 0.01 vs. DAT^+/+^ rats.

**Figure 3 biomolecules-13-00516-f003:**
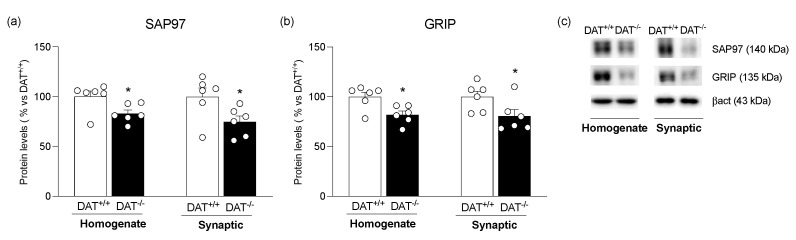
Protein expression level of specific AMPA receptor anchoring proteins SAP97 (**panel a**) and GRIP (**panel b**) in the mPFC of DAT^+/+^ and DAT^−/−^ rats. In (**panel c**), representative immunoblots for SAP97, GRIP, and β-Actin are shown for each fraction. Results obtained in the whole homogenate and in the synaptic fraction are expressed as mean percentage ± mean standard error from six independent determinations for each experimental group. Unpaired Student’s *t*-test * *p* < 0.05 vs. DAT+/+ rats.

**Figure 4 biomolecules-13-00516-f004:**
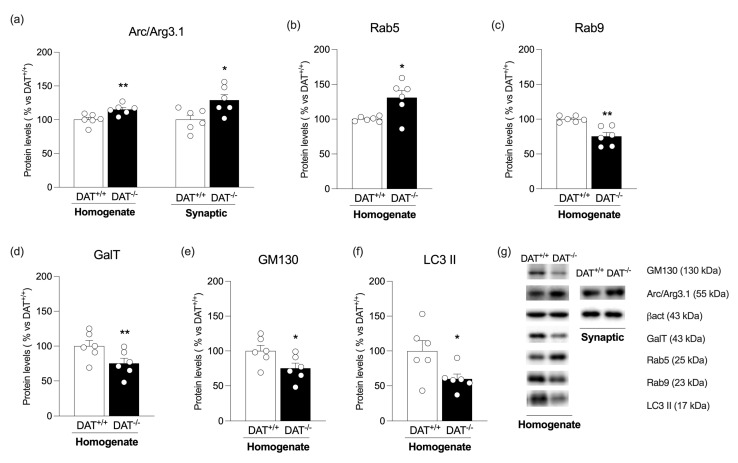
Protein expression level of endosomal and autophagy markers in the mPFC. Arc/Arg 3.1 (**panel a**) was investigated in the whole homogenate and in the PSD of DAT^+/+^ and DAT^−/−^ rats. Rab5 (**panel b**), Rab9 (**panel c**), GalT (**panel d**), GM130 (**panel e**), and LC3-II (**panel f**) were investigated in the whole homogenate of DAT^+/+^ and DAT^−/−^ rats. In (**panel g**), representative immunoblots are shown for Arc/Arg3.1, Rab5, Rab9, GalT, GM130, LC3-II, and β-Actin. Results are expressed as mean percentage ± mean standard error from six independent determinations for each experimental group. Unpaired Student’s *t*-test * *p* < 0.05, ** *p* < 0.01 vs. DAT^+/+^ rats.

**Figure 5 biomolecules-13-00516-f005:**
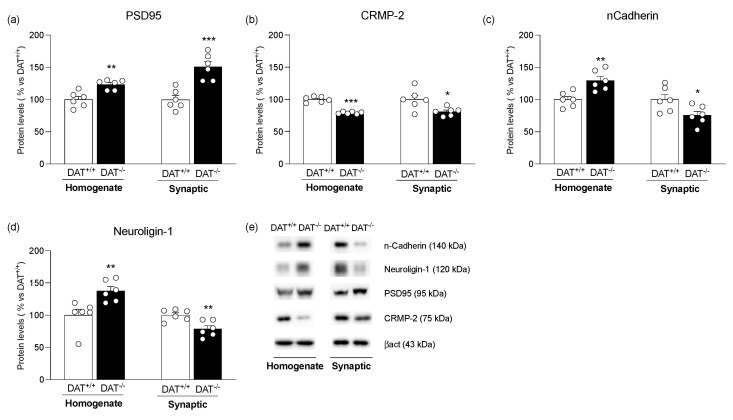
Protein expression level of structural markers in the mPFC of DAT^+/+^ and DAT^−/−^ rats. PSD95 (**panel a**), CRMP-2 (**panel b**), nCadherin (**panel c**), and Neuroligin-1 (**panel d**) were measured in the whole homogenate and in the synaptic fraction. In (**panel e**), representative immunoblots are shown for PSD95, CRMP-2, nCadherin, Neuroligin-1, and β-Actin. Results are expressed as mean percentage ± mean standard error from six independent determinations for each experimental group. Unpaired Student’s *t*-test * *p* < 0.05, ** *p* < 0.01, *** *p* < 0.001 vs. DAT^+/+^ rats.

**Figure 6 biomolecules-13-00516-f006:**
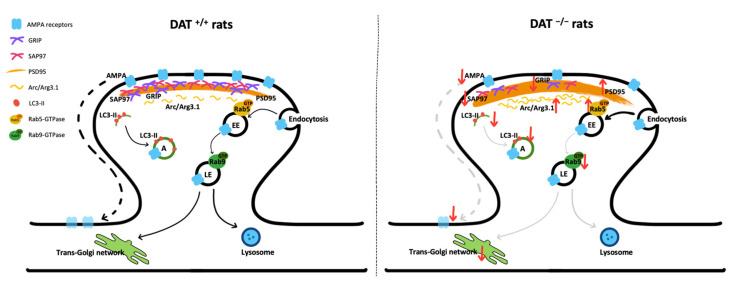
Schematic representation of the changes observed in the glutamatergic synapse in the PFC of DAT^−/−^ rats. The condition of hyperdopaminergia caused by DAT deletion alters the trafficking of AMPA receptors and their scaffolding protein expression. Changes in the endosomal and autophagic mechanisms are paralleled with the impairment of the glutamatergic synapse. *AMPA*, glutamate-amino-3-hydroxy-5-methyl-4-isoxazole propionic acid receptors; *SAP97,* synapse-associated protein 97; *GRIP,* glutamate receptor-interacting protein; *PSD95.* postsynaptic density protein 95; *Arc/Arg3.1,* Activity-regulated cytoskeleton-associated protein; *Rab5,* Ras-associated binding protein 5; *Rab9,* Ras-associated binding protein 9; *LC3-II,* microtubule-associated proteins 1A/1B light chain 3B; *EE,* early endosome; *LE,* late endosome; *A,* autophagosome.

## Data Availability

The data that support the findings of this study are available from the corresponding author upon reasonable request.

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
