# Peer review of "Dysregulation of AMPA Receptor Trafficking and Intracellular Vesicular Sorting in the Prefrontal Cortex of Dopamine Transporter Knock-Out Rats"

_biomolecules, 2023, doi:10.3390/biom13030516_

Round 1

Reviewer 1 Report

The study with title: Dysregulation of AMPA receptor trafficking and intracellular vesicular sorting in the prefrontal cortex of dopamine transporter knock-out rats " show that  DA overactivity alters the homeostatic plasticity of the glutamate  synapse in e in the mPFC through profound dysregulation of AMPA receptor trafficking, an effect likely involving increased internalization of such receptors and alterations in endosomal trafficking and degradation.

correct the English throughout the manuscript

Author Response

Reviewer 1. The study with title: Dysregulation of AMPA receptor trafficking and intracellular vesicular sorting in the prefrontal cortex of dopamine transporter knock-out rats " show that  DA overactivity alters the homeostatic plasticity of the glutamate  synapse in e in the mPFC through profound dysregulation of AMPA receptor trafficking, an effect likely involving increased internalization of such receptors and alterations in endosomal trafficking and degradation.

  • correct the English throughout the manuscript

Answer: We thank reviewer 1 who likes the manuscript in its present form. To accomplish the reviewer’s request, we have double checked the manuscript to correct for English mistakes.

Reviewer 2 Report

Giorgia Targa and coworkers report that the traffic and vesicular sorting of AMPA receptors is affected in the brain of rats knock-out  for the dopamine transporter (DAT).

The authors have prepared synaptic fractions from WT and DAT-/- rats. They have analysed by western-blot the expression of AMPA R subunits, together with proteins known to be involved in their trafficking. To have an idea about the subcellular localisation of AMPA receptors, they have used the whole homogenate, the synaptic and the extra-synaptic fractions.

Overall, the study was well conducted. However, I still have some questions/concerns to improve this study:

1/ Is the stochiometry of AMPA R heteroduplexes affected in DAT KO animals ? To answer this question, I suggest that authors perform co-immunoprecipitation between the different AMPA R subunits.

2/ Similarly, is their interaction with scaffolding proteins (PSD95…) affected?

3/ Do the receptors are at the surface?

4/ Is their function affected in DAT-/- animals (patch-clamp)? This will complement other data that were all obtained by western blots.

5/ The authors claim that this is the translation of receptors that are affected but they don’t show it. They show that there are fewer proteins for receptors but this could be due to many events: are the AMPAR coding genes transcribed at the same level in WT vs KO? (RT-qPCR for Gria1 to Gria4). Are the receptors less stable, and more targeted to the destruction machinery (polyubiquitinated)?

6/ The authors use the DAT KO as a model of DA overactivity, which is logic. However, they never consider that the DAT may have some interactors and that in the absence of DAT, their interactors’ function might change, with some consequences on synaptic activity independent of dopamine and the trafficking of AMPA receptors. The DAT is known to interact with many proteins (see Dopamine transporter/syntaxin 1A interactions regulate transporter channel activity and dopaminergic synaptic transmission, by Cravelli et al, PNAS 2008). A  list of interactors can be found in Regulation of dopamine transporter function by protein-protein interactions: discoveries and methodological challenges, by Eriksen et al, JNC, 2010. Thus, it might be that what the authors observed is due to DA hyperactivity and/or change in DAT interactors functions. At least the authors should mention this possibility. Better, to tackle this issue, authors could treat WT animals with a dopamine-releasing agent. In this condition, there should be more DA at the synapse.

Author Response

Reviewer 2. Giorgia Targa and coworkers report that the traffic and vesicular sorting of AMPA receptors is affected in the brain of rat knock-out for the dopamine transporter (DAT). The authors have prepared synaptic fractions from WT and DAT-/- rats. They have analyzed by western blot the expression of AMPA R subunits, together with proteins known to be involved in their trafficking. To have an idea about the subcellular localization of AMPA receptors, they have used the whole homogenate, the synaptic and the extra-synaptic fractions.

Overall, the study was well conducted. However, I still have some questions/concerns to improve this study:

1/ Is the stochiometry of AMPA R heteroduplexes affected in DAT KO animals? To answer this question, I suggest that authors perform co-immunoprecipitation between the different AMPA R subunits.

Answer: We thank the reviewer for raising this interesting point, since AMPAR subunit combinations have not been systematically analyzed throughout the brain. In the mPFC of naïve rat, evaluation of the subunit composition of AMPA receptor populations indicates a major role for GluA1/A2 receptors and only a minor fraction is represented by membrane GluA2/A3 receptors and GluA2-lacking AMPA receptors (Reimers et al., 2011). In the same paper, authors suggest that co-IP experiments measure the total cellular pool of AMPA receptors without selectively distinguish AMPA receptor in the synaptic membrane from the intracellular pool and partially assembled AMPA in the endoplasmic reticulum. Since we are interested in evaluating DA influence on AMPA receptors localized in the active site of the synapse, performing co-IP experiments goes beyond the scope of our manuscript. Nevertheless, we here show that DAT deletion reduced GluA1 and GluA2 while increased GluA3 protein levels in the homogenate, whereas all three subunits were decreased in the post-synaptic density. These changes reveal a DA-induced redistribution of the AMPA subunit expression (thus favoring GluA3 translation) and consequently an altered AMPA receptor recruitment at the synapse (reducing GluA1/GluA2 heterodimer), suggesting that DA shapes cortical synaptic strength.

2/ Similarly, is their interaction with scaffolding proteins (PSD95…) affected?

Answer: We thank the reviewer for raising this point. We had not measured the interaction of the major AMPA scaffolding proteins (GRIP, SAP97) with their respective AMPA receptors; however, since we have a purified preparation of the post-synaptic density, we believe that the reductions of both GluA1/SAP97 and GluA2/GRIP are likely indicative of an interaction.

3/ Do the receptors are at the surface? 

Answer: As we have explained in the manuscript, we made a purified preparation of the post-synaptic density (triton insoluble fraction), which is defined as a protein dense specialization localized at the post-synaptic sites of excitatory synapses where the two main types of glutamate-receptor channels are clustered. Accordingly, we cannot rule out the possibility that the herein observed effects are due also to recycling vesicles stored in close vicinity to the membrane; however the reduced expression of GuA1 and GluA2 observed in the post-synaptic density in concomitance with the increased expression of the early endosome markers Rab5 clearly suggest that DA overactivity destabilizes the glutamate synapse reducing AMPA availability at both synaptic and extra-synaptic sites. Since we did not measure AMPA receptors specifically at the surface, we have toned down the following sentence that now reads (Pag 9 line 293): ‘It appears, thus, that a physiological concentration of DA is critical for the maintenance of AMPA receptor trafficking at the PSD’.

4/ Is their function affected in DAT-/- animals (patch-clamp)? This will complement other data that were all obtained by western blots.

Answer: We agree with reviewer 2 with respect to this comment and we are aware that electrophysiological studies would add important information to our suggestion of ‘altered homeostatic plasticity of the glutamate synapse’. However, we have not run patch clamp experiments to measure whether AMPA receptors are functional. A study conducted by Bai et al. 2014 demonstrated through extracellular field recording that GBR 12909 and GBR12935, specific DAT inhibitors, were able to impair physiological early LTD leading to a diminished plasticity induction in the rat mPFC. We have added this evidence in the manuscript (pag 9 lines 326-329): ‘Our hypothesis is in line with previous data showing that specific DAT inhibitors and elevated DA levels in cortical slices from naïve rats impaired long-term depression, thus inducing abnormal PFC activity (Bai et al., 2014)’.

5/ The authors claim that this is the translation of receptors that are affected but they don’t show it. They show that there are fewer proteins for receptors, but this could be due to many events: are the AMPAR coding genes transcribed at the same level in WT vs KO? (RT-qPCR for Gria1 to Gria4). Are the receptors less stable, and more targeted to the destruction machinery (polyubiquitinated)?

Answer: We thank the reviewer for his/her comment that allows us to better clarify the interpretation of the data obtained in the whole homogenate. When we use the term ‘translation’ we are referring to the fact that, given the reduced levels of GuA1 and GluA2 in the homogenate, the increased dopaminergic tone reduced AMPA receptor subunit synthesis. Of course, such reduced translation could be due to several reasons. For instance, it could be due to reduced transcription of the different subunits (as suggested by the reviewer) but it could also be due to the modulatory effect of miRNA or to post-translational changes, such as SUMOylation or ubiquitination. In this manuscript we focused our attention on the synaptic retention and trafficking of AMPA receptors between synaptic and extra-synaptic sites and to the mechanisms that may contribute to explain such altered trafficking. In this scenario, the investigation of the mechanisms responsible of down-regulation of GluA1 and GluA2 synthesis goes beyond our interest and need specific investigations. We discuss this point pag 9 lines 319-321 and the sentence reads: ‘The reduced GluA1 and GluA2 translation might be due to reduced transcription of the different subunits, to the modulatory effect of miRNA or post-translational changes, such as SUMOylation or ubiquitination (Corti and Duarte, 2023)’.

6/ The authors use the DAT KO as a model of DA overactivity, which is logic. However, they never consider that the DAT may have some interactors and that in the absence of DAT, their interactors’ function might change, with some consequences on synaptic activity independent of dopamine and the trafficking of AMPA receptors. The DAT is known to interact with many proteins (see Dopamine transporter/syntaxin 1A interactions regulate transporter channel activity and dopaminergic synaptic transmission, by Cravelli et al, PNAS 2008). A list of interactors can be found in Regulation of dopamine transporter function by protein-protein interactions: discoveries and methodological challenges, by Eriksen et al, JNC, 2010. Thus, it might be that what the authors observed is due to DA hyperactivity and/or change in DAT interactors functions. At least the authors should mention this possibility. Better, to tackle this issue, authors could treat WT animals with a dopamine-releasing agent. In this condition, there should be more DA at the synapse.

Answer: We thank reviewer #1 for raising this point and we agree with him/her that while the majority of data indicate that the observed changes are related to overactive DA system in the prefrontal cortex, secondary to disrupted re-uptake of DA, the possibility of disrupted DAT-dependent protein-protein interaction leading to alterations in the function of other proteins known to be interacting with the DAT (Cravelli et al, PNAS 2008; Eriksen et al, JNC, 2010) cannot be fully excluded. We have added this concept in the Discussion (pag. 11, lines 498-501), the sentence now reads: ‘In addition, since DAT functions may be regulated also by protein-protein interactions (Carvelli et al, 2008; Eriksen et al, 2010), we cannot rule out the possibility that change in DAT interactors functions may contribute to the alteration of AMPA receptor trafficking at the PSD’.

Reviewer 3 Report

I have only one major comment concerning the results. Authors use knock-out (DAT-/-) rats to produce stable DA accumulation in extracellular space in the prefrontal cortex. Authors suggest, but do not measure extracellular DA accumulation in the prefrontal cortex. However there are data in mice that DAT -/- genotype do not result in significant extracellular DA elevation in prefrontal cortex (Shen et al., 2004), while acute DAT inhibition with cocaine do. Authors show numerous distortions in proteins expression, which may be explained by DA hyperactivity in prefrontal cortex, but the question of actual DA concentration in PFC in DAT-KO mice is not clear. This contradiction needs an explanation.

Shen HW, Hagino Y, Kobayashi H, Shinohara-Tanaka K, Ikeda K, Yamamoto H, Yamamoto T, Lesch KP, Murphy DL, Hall FS, Uhl GR, Sora I. Regional differences in extracellular dopamine and serotonin assessed by in vivo microdialysis in mice lacking dopamine and/or serotonin transporters. Neuropsychopharmacology. 2004 Oct;29(10):1790-9. doi: 10.1038/sj.npp.1300476. PMID: 15226739.

Minor comment: Norepinephrine (NET) transporter has a significant role in DA uptake (Morón et al., 2002) and can be proposed as functional DAT replacement in DAT-KO animals. This needs to be discussed.

Morón JA, Brockington A, Wise RA, Rocha BA, Hope BT. Dopamine uptake through the norepinephrine transporter in brain regions with low levels of the dopamine transporter: evidence from knock-out mouse lines. J Neurosci. 2002 Jan 15;22(2):389-95. doi: 10.1523/JNEUROSCI.22-02-00389.2002. PMID: 11784783; PMCID: PMC6758674.

Author Response

Reviewer 3. I have only one major comment concerning the results. Authors use knock-out (DAT-/-) rats to produce stable DA accumulation in extracellular space in the prefrontal cortex. Authors suggest, but do not measure extracellular DA accumulation in the prefrontal cortex. However, there are data in mice that DAT -/- genotype do not result in significant extracellular DA elevation in prefrontal cortex (Shen et al., 2004), while acute DAT inhibition with cocaine do. Authors show numerous distortions in proteins expression, which may be explained by DA hyperactivity in prefrontal cortex, but the question of actual DA concentration in PFC in DAT-KO mice is not clear. This contradiction needs an explanation.

Shen HW, Hagino Y, Kobayashi H, Shinohara-Tanaka K, Ikeda K, Yamamoto H, Yamamoto T, Lesch KP, Murphy DL, Hall FS, Uhl GR, Sora I. Regional differences in extracellular dopamine and serotonin assessed by in vivo microdialysis in mice lacking dopamine and/or serotonin transporters. Neuropsychopharmacology. 2004 Oct;29(10):1790-9. doi: 10.1038/sj.npp.1300476. PMID: 15226739.

Answer: The reviewer mentions the evidence from Shen and associates (2004) who did not show differences in the extracellular levels of dopamine in the prefrontal cortex of DAT KO mice, using in vivo microdialysis. However, to the best of our knowledge, another evidence exists from Xu and colleagues (2009) showing a 3-fold increase in the prefrontal cortex. In this manuscript the authors used quantitative low perfusion rate microdialysis (flow rate was 0.1mL/min) at variance from Shen and associates who used regular microdialysis (1ml/min) that may not detect relatively small changes such as the 3-fold increase shown by Xu and colleagues. The need to use quantitative microdialysis approaches (like the low perfusion rate microdialysis) to accurately measure baseline is nicely explained in the manuscript from Parsons and Justice (Crit Rev Neurobiol; 8:189-220, 1994). In addition, we also have to take into account that striatal and accumbal DA may spill over to neighboring regions, thus elevating DA levels in regions that normally have low levels of extracellular DA, and this is likely if we consider the 5-fold increase of dopamine levels in DAT KO mice striatum. Based on these considerations, we assume that the removal of DAT has caused a significant increase in the extracellular levels of dopamine in the prefrontal cortex of DAT KO rats that, although not similar in magnitude to the increase observed in striatum (Leo et al., 2018), it may indeed influence the homeostasis of other neurotransmitters, such as glutamate as shown in our manuscript.

We have added this concept in the introduction (pag. 2, lines 89-92), the sentence now reads: ‘Based on the above-mentioned evidence, our goal was to investigate if and how hyperactivity of the DAergic system at the prefrontal cortex level, shown by previous work in DAT-/- mice (Xu et al., 2009), would influence AMPA subunit composition, trafficking and localization (Xu et al., 2009).’

Minor comment: Norepinephrine (NET) transporter has a significant role in DA uptake (Morón et al., 2002) and can be proposed as functional DAT replacement in DAT-KO animals. This needs to be discussed.

Morón JA, Brockington A, Wise RA, Rocha BA, Hope BT. Dopamine uptake through the norepinephrine transporter in brain regions with low levels of the dopamine transporter: evidence from knock-out mouse lines. J Neurosci. 2002 Jan 15;22(2):389-95. doi: 10.1523/JNEUROSCI.22-02-00389.2002. PMID: 11784783; PMCID: PMC6758674.

Answer: We thank the reviewer for raising this point. We are aware that NET has a relevant role in DA clearance from the mPFC in concomitance with DAT activity. For instance, evidence from Carboni et al. 2006 shows that co-administration in naive rats of GBR 12909, a selective inhibitor of DAT, and reboxetine, a selective blocker of the norepinephrine transporter (NET), causes a larger increment in dialysate DA in the medial prefrontal cortex than the sum of the two compounds alone. Thus, based on the data in the literature, we cannot exclude a role for NET in cortical DA reuptake and consequently in the changes that we have observed in the prefrontal cortex of DAT-/- rats. In addition, as mentioned above, we also have to take into account the potential spillover of dopamine from the striatum where there is an increase of 5-fold of its extracellular levels.

As suggested, we have implemented the discussion (pag. 11, line 420-424) adding the following sentence ‘We are also aware that the norepinephrine transporter may come into play as functional DAT replacement in the prefrontal cortex of DAT-/- rats (Moron et al., 2002; Carboni et al., 2006). However, evidence exists that extracellular levels of DA are significantly enhanced in the PFC of DAT-/- mice (Xu et al., 2009), potentially ruling out the possibility that NET may have prevented cortical DA hyperactivity

Round 2

Reviewer 2 Report

The authors have satisfactorily addressed my comments.